# Grave-to-cradle photothermal upcycling of waste polyesters over spent LiCoO$_2$

Xiangxi Lou[1,2,5], Penglei Yan[1,5], Binglei Jiao[3,5], Qingye Li[1], Panpan Xu[3,6] ✉,
Lei Wang[1], Liang Zhang[1], Muhan Cao[1], Guiling Wang[2], Zheng Chen[4],
Qiao Zhang[1] & Jinxing Chen[1,6] ✉

Lithium-ion batteries (LIBs) and plastics are pivotal components of modern society; nevertheless, their escalating production poses formidable challenges to resource sustainability and ecosystem integrity. Here, we showcase the transformation of spent lithium cobalt oxide (LCO) cathodes into photothermal catalysts capable of catalyzing the upcycling of diverse waste polyesters into high-value monomers. The distinctive Li deficiency in spent LCO induces a contraction in the Co−O$_6$ unit cell, boosting the monomer yield exceeding that of pristine LCO by a factor of 10.24. A comprehensive life-cycle assessment underscores the economic viability of utilizing spent LCO as a photothermal catalyst, yielding returns of 129.6 \$·kg$_{LCO}^{-1}$, surpassing traditional battery recycling returns (13–17 \$·kg$_{LCO}^{-1}$). Solar-driven recycling 100,000 tons of PET can reduce $3.459 \times 10^{11}$ kJ of electric energy and decrease 38,716 tons of greenhouse gas emissions. This work unveils a sustainable solution for the management of spent LIBs and plastics.

The transition in the global energy landscape, moving away from conventional fossil fuels toward renewable energy sources, has underscored the imperative need for dependable technologies for efficient energy conversion and storage[1,2]. Among these technologies, Lithium-ion batteries (LIBs) have emerged as pivotal components within the burgeoning electric vehicle and stationary energy storage grid markets[3,4]. While LIBs have sparked transformative advancements in the energy sector, an accompanying challenge is that millions of metric tons of critical metal resources become embedded within municipal waste streams following the failure of LIBs[5,6]. This mounting volume of spent LIBs poses formidable obstacles to waste management and environmental conservation. In contrast to conventional pyrometallurgy and hydrometallurgy, as well as regeneration approaches employed in LIB recycling, the conversion of spent LIBs into advanced functional materials offers a promising alternative[7–9]. This strategy circumvents the energy-intensive yet often inefficient phases of "recovery and purification", thus aligning with higher echelons of the waste management hierarchy[10–12].

In recent years, the widespread global utilization of plastic products has precipitated a continual escalation in the annual accumulation of plastic waste, thereby engendering detrimental repercussions for the environment[13–18]. The emergence of solar-driven photothermal plastic upcycling as a paradigm holds substantial promise in the realms of sustainable plastic reutilization[19–24]. While the development of photothermal catalysts that possess dual attributes of robust sunlight absorption and elevated catalytic activity is crucial[25], recognizing sustainability as an additional dimension of photothermal catalysts through uncomplicated modifications using discarded LIBs represents a groundbreaking paradigm shift in upcycling. This innovative approach transcends disciplinary boundaries, concurrently enhancing energy efficiencies in waste recycling while advancing the principles of circular economy.

[1]Institute of Functional Nano & Soft Materials (FUNSOM), Jiangsu Key Laboratory for Carbon-Based Functional Materials & Devices, Soochow University, Suzhou 215123 Jiangsu, China. [2]Key Laboratory of Superlight Materials and Surface Technology of Ministry of Education, College of Materials Science and Chemical Engineering, Harbin Engineering University, Harbin 150001 heilongjiang, China. [3]Advanced Materials Division, Suzhou Institute of Nano-Tech and Nano-Bionics, Chinese Academy of Sciences, Suzhou 215123 Jiangsu, China. [4]Department of NanoEngineering, University of California San Diego, La Jolla 92093 CA, USA. [5]These authors contributed equally: Xiangxi Lou, Penglei Yan, Binglei Jiao. [6]These authors jointly supervised this work: Panpan Xu, Jinxing Chen. ✉e-mail: panpanxu2021@sinano.ac.cn; chenjinxing@suda.edu.cn

In this work, we present a novel approach for upgrading spent LIBs, wherein the extraction of battery cathode material serves as a photothermal catalyst for various waste polyester recycling processes. Under simulated sunlight irradiation of 0.82 W cm$^{-2}$ for 30 min, the conversion rate of PET reaches 96.34%, with the purity of the resulting product bis(2-hydroxyethyl) terephthalate (BHET) exceeding 99.5%. Additionally, the catalyst exhibits excellent stability over extended reaction times (> 70 h). Further outdoor experiments, energy consumption assessments, and environmental impact analyses all underscore the effectiveness and eco-friendliness of PET and retired battery recycling.

## Results

### Upgrading spent lithium cobalt oxide cathodes to catalysts

The production of lithium cobalt oxide (LCO) and other cobalt-containing battery materials currently accounts for more than 70% of the world's cobalt resource consumption in 2022[26]. While the LIB industry is actively working to minimize cobalt usage, it is crucial to acknowledge that cobalt plays a vital role in various applications, particularly in catalysis. Recycling cobalt resources for use in other applications can help mitigate the need for additional cobalt mining and refining activities, significantly contributing to reducing the $CO_2$ footprint. In this context, we used discarded LCO as a catalyst for polyester upcycling, addressing environmental concerns while also enhancing the sustainability of critical materials. Our dedicated efforts are centered on diversifying possible applications for cathode materials derived from spent LIBs[27,28].

To elucidate the pivotal role of waste LCO in the reutilization of photothermal catalytic plastics, we have conducted a comprehensive life cycle assessment. The revenue calculation was based on the sales of recycled materials based on EverBatt model, developed by Argonne National Laboratory[29]. As illustrated in Fig. 1a, b, traditional recycling pathways for waste LCO material are associated with profits ranging from 13 to 17 \$·kg$_{LCO}^{-1}$. In stark contrast, repurposing and upgrading these spent LCO materials for utilization as catalysts in the recycling of polyester plastics can yield profits of 64.6 \$·kg$_{LCO}^{-1}$. By extending the replacement interval of the catalysts from biweekly to monthly, the profit escalates to 129.6 \$·kg$_{LCO}^{-1}$. Consequently, the conversion and recycling of aged LCO batteries into catalysts emerge as an exceedingly enticing and economically viable proposition. Figure 1c outlined the upgrading spent LCO into photothermal catalysts for waste polyester upcycling.

### Controlled chemical delithiation of LCO

It is imperative to acknowledge that LCO materials, when sourced from various retired states, may exhibit distinct structural configurations. Establishing a definitive correlation between spent LCO and its efficacy in catalyzing the upgrading of polyester poses a formidable challenge. To this end, we conducted a controllable chemical delithiation procedure to mimic the Li$^+$ depletion in charge-discharge cycles. Prior to the chemical delithiation, the pristine LCO was subjected to ball milling to reduce the catalyst's particle size to around 1 micron. Subsequently, we employed a potassium persulfate (KPS) solution to oxidize the LCO with different times, thus partially removing Li$^+$ (Fig. 2a)[30]. The Co and Li contents in each sample were quantified by inductively coupled plasma optical emission spectroscopy (ICP-OES). As shown in Fig. 2b and Supplementary Table 1, neither the pretreatment nor the chemical delithiation process resulted in any discernible loss of Co, while Li was removed from powders. The relationship between chemical delithiation time and delithiation degree is shown in Supplementary Fig. 1. Utilizing the Li/Co ratios as reference, we designated names for the five samples: LiCoO$_2$ (pristine), Li$_{0.92}$CoO$_2$ (ball milling), Li$_{0.76}$CoO$_2$ (4 h), Li$_{0.38}$CoO$_2$ (8 h), and Li$_{0.25}$CoO$_2$ (12 h), respectively (Supplementary Figs. 2–5). Furthermore, we implemented an electrochemical delithiation method to regulate the lithiation degree of LCO[31]. Initially, the

assembled LCO coin cells were charged at a rate of 0.1 C with a cutoff voltage of 4.3 V (Supplementary Fig. 6). The lithiation degree of the charged LCO cathode was determined to be 0.56 by ICP-OES measurement. The correlation between charging voltage and lithiation degree is illustrated on the upper x-axis in Supplementary Fig. 6a. To achieve specific lithiation degrees of 0.92, 0.76, and 0.60 for LCO, we set cutoff voltages of 3.93, 3.97, and 4.08 V (Supplementary Fig. 6b–d). The lithium contents measured in the charged LCO cathodes closely align with the predefined values (Supplementary Table 2).

The X-ray diffraction (XRD) patterns of the various catalysts provide confirmation that Li$_{0.92}$CoO$_2$ and Li$_{0.76}$CoO$_2$ retained the characteristic crystalline structure of LiCoO$_2$ (Supplementary Fig. 7). To gain deep insights into the impact of chemical delithiation on the crystal structure, we performed Rietveld refinement on the XRD patterns of both the pristine LCO (Fig. 2c) and Li$_{0.76}$CoO$_2$ (Fig. 2d). The lattice parameters of the LCO and Li$_{0.76}$CoO$_2$ powders were calculated using the least squares method and are listed in Supplementary Table 3. The decreased lattice constant $a$ and increased lattice constant $c$ after delithiation both suggest that Li$_{0.76}$CoO$_2$ had a weakening of the shielding effect of O$^{2-}$ on Li$^+$, resulting in increased repulsion between oxygen layers and expansion along the c-axis[32]. Conversely, Li$_{0.25}$CoO$_2$ suffered from a substantial structure destroy. This observation suggests that prolonged delithiation induces structural disorder within the material.

To gain further insight into changes in bonding characteristics and local structure, we conducted Co K-edge X-ray absorption near-edge structure (XANES) analysis of Li$_{1-x}$CoO$_2$ (Fig. 2e). The shift of the edge at 7709 eV towards higher energy during delithiation process indicates an increase in the binding energy of core electrons, suggesting the oxidation of Co species[33]. The pre-edge peak at 7719 eV represents the electric dipole-allowed $1s \rightarrow 4p$ transition (Fig. 2f). Throughout the chemical delithiation process, the disappearance of this transition aligns with an increase in ionicity, signifying a reduced overlap between O $2p$ orbitals and Co $3d - 4p$ hybridized orbitals. The high-intensity peak at ~7728 eV, with edge energy being linked to the oxidation state of Co ions and the local electronic structure, demonstrates variations in edge energy (Fig. 2g). As the chemical delithiation process unfolds, the edge of Li$_{1-x}$CoO$_2$ shifts from ~7728 eV to ~7730 eV, indicating an elevated binding energy of Co ions in Li$_{0.76}$CoO$_2$ relative to LiCoO$_2$. Figure 2h displays Fourier transforms of k$^3$-weighted Co K-edge extended X-ray absorption fine structure (EXAFS) spectra. In both EXAFS spectra, two prominent peaks emerge, centered at phase-uncorrected radial distances of approximately ~1.47 and ~2.42 Å, which are attributed to the nearest Co−O and Co − (O)−Co bond distances, respectively[34]. During the chemical delithiation process, the diminishing intensity of the Co−O peak and the intensifying of Co−Co peak both indicate distortion of the Co−O$_6$ octahedra and oxidation of Co ions. Figure 2i summarizes the structural evolution during delithiation process of LiCoO$_2$. With the removal of Li$^+$, the electrostatic repulsion between Co−O$_6$ layers increases, which leads to an increase in the interlayer distance ($d'_2 > d_2$). At the same time, the removal of Li$^+$ also causes an increase in the valence state of Co, increasing the interaction within the Co−O$_6$ unit cell. Accordingly, the shortening of Co−O bond reduces the thickness of Co−O$_6$ layers layer ($d'_1 < d_1$).

### Structure-performance relationship

Since polyethylene terephthalate (PET) is the most widely used polyester plastic, we employed photothermal-catalyzed glycolysis of PET to BHET as the model reaction to evaluate catalytic performance of different catalysts. Attaining superior photothermal conversion efficiency stands as an essential prerequisite in the domain of photothermal catalysis. We conducted an evaluation of the light absorption capabilities of various catalysts, which all exhibited pronounced light absorption capabilities (Supplementary Fig. 8). With a mass loading of

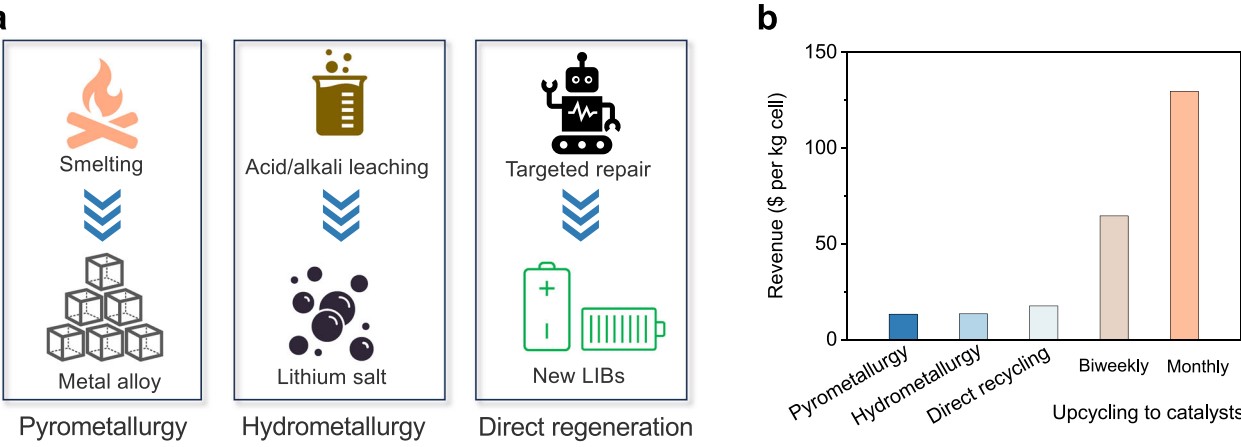

**Fig. 1 | Upgrading spent LCO cathodes to catalysts. a** Traditional LCO battery recycling strategies, including pyrometallurgy, hydrometallurgy and direct regeneration. **b** Economic benefits of different strategies for LCO cathodes recycling. **c** Schematic diagram of the recovery process of spent LCO and photothermal catalytic polyester recycling. Initially, the discarded LCO batteries were disassembled to acquire photothermal catalysts. Subsequently, these catalysts were combined with shredded PET film and ethylene glycol (EG). The resultant solution was then subjected to simulated sunlight to initiate the reaction. Finally, a sequence of separation and purification steps was employed to purify BHET.

0.4 wt.% in ethylene glycol (EG) solution, the temperature escalated to 190 °C within a mere 30 min under simulated sunlight conditions (0.82 W cm⁻², Supplementary Fig. 9). Importantly, the temperature elevation profiles for different catalysts closely converged, underscoring LCO's exceptional prowess in converting light to heat. To ascertain its photothermal stability, we subjected LCO to continuous heating-cooling cycles lasting over 4 h, and the results evinced no performance deterioration (Supplementary Fig. 10). The impact of delithiation degree on catalytic performance is presented in Fig. 3a, revealing a distinctive volcano-shaped trend. In comparison to pristine LCO, delithiated LCO exhibited enhanced catalytic activity. For instance, when $Li_{0.76}CoO_2$ was employed as a catalyst, its yield of BHET surpassed that of pristine LCO by a factor of 10.24. The XANES spectra of these delitiated LCO indicate that delithiation of LCO results in an increase in the valence state of the Co element ($Co^{δ+}$, δ > 3). This alteration in the valence state induces the contraction of the $Co-O_6$ unit cell, subsequently enhancing the efficiency of $Co^{δ+}$ in activating carbonyl oxygen in PET. In contrast, in $LiCoO_2$, where Co exists

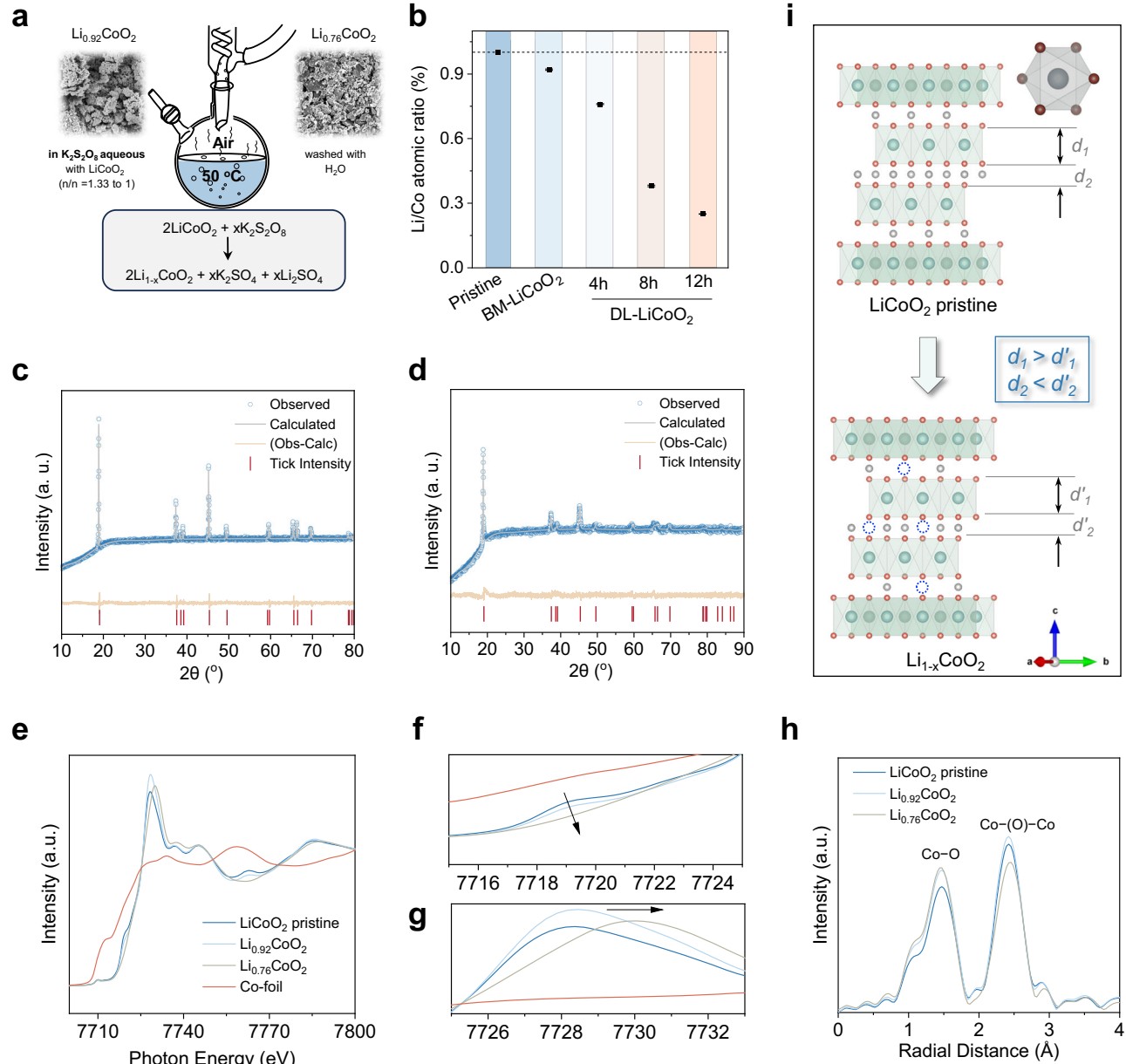

**Fig. 2 | Catalyst characterization. a** Reaction scheme for the chemical delithiation of LCO. **b** Li/Co (mol/mol) ratio in various samples, including LCO pristine, treated with ball milling for 4 h (BM-LiCoO$_2$), chemical delithiation with 4 h, 8 h and 12 h. **c**, **d** Rietveld refinement powder XRD patterns of LiCoO$_2$ pristine (**c**) Li$_{0.76}$CoO$_2$ (**d**).

**e**–**g** Co K-edge XANES spectra of LiCoO$_2$, Li$_{0.92}$CoO$_2$, Li$_{0.76}$CoO$_2$ and Co foil. **h** k$^3$-weighted Fourier-transformed EXAFS of LiCoO$_2$, Li$_{0.92}$CoO$_2$, and Li$_{0.76}$CoO$_2$. **i** Schematic views of the atomic structures of LiCoO$_2$ and Li$_{1-x}$CoO$_2$. The green, red, and gray spheres represent Co, O and Li atoms, respectively.

predominantly in the form of Co$^{3+}$, its ability to activate carbonyl oxygen atoms in the PET chain is comparatively weak. However, as the degree of delithiation increased, the structural integrity of LCO was compromised, leading to reduced catalytic activity. Subsequently, we explored the electrochemically delithiated LCO samples for catalyzing PET glycolysis (Supplementary Fig. 11). Our findings revealed that the catalytic activity was most pronounced for the LCO sample with a lithiation degree of 0.72, aligning with the results observed in chemically delithiated samples. This outcome underscores the effectiveness of the electrochemical delithiation method in regulating the lithiation degree.

Based on the above structure-performance relation, we suggest that achieving an optimal lithium-cobalt ratio for retired materials is important. When the lithiation degree in spent LIBs falls below the optimal value, a relithiation step might be necessary. In the field of

battery recycling, relithiation technologies have seen significant progress, with various methods developed, including solid-state relithiation, chemical relithiation, ionothermal relithiation, hydrothermal relithiation, and molten salt relithiation. For obtaining LCO with a predefined Li composition, solid-state relithiation with precise control of Li salt addition would be preferable. This process is straightforward and holds great potential for practical applications.

We further revealed the effect of the localized heating effect on photothermal catalysis. As shown in Fig. 3b, subsequent to catalytic PET glycolysis conducted at 190 °C for 10 min, photothermal catalysis achieved a PET conversion rate of 39.88% and a BHET yield of 18.66%. These figures surpassed those obtained through thermal catalysis by a remarkable factor of 3.1 (12.96%) and 8.7 (2.13%) respectively, all under identical reaction conditions. In an effort to elucidate the underlying mechanism, ultraviolet (UV) light was applied to the thermal catalytic

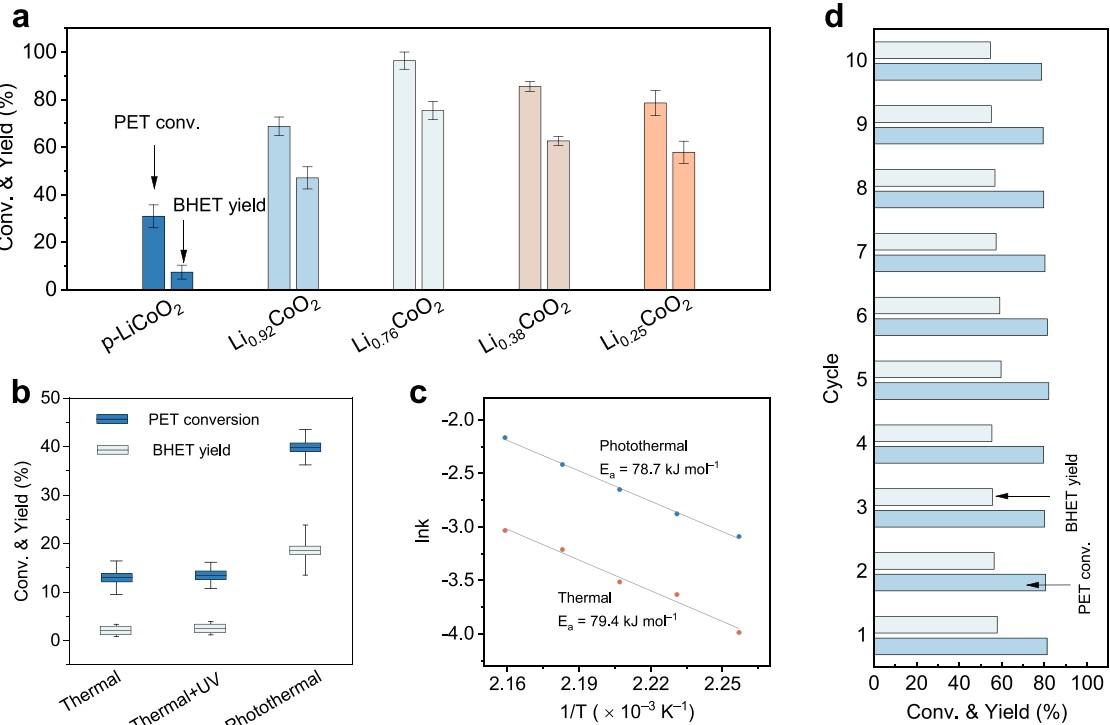

**Fig. 3 | Photothermal catalytic PET glycolysis over $Li_{1-x}CoO_2$. a** Conversion of PET and yield of BHET over $Li_{1-x}CoO_2$. **b** Effect of UV light (0.01 W cm⁻²) on the depolymerization of PET in photothermal catalysis and thermal catalysis. **c** Arrhenius plot of the rate constant of PET glycolysis for $Li_{0.76}CoO_2$ through thermal and photothermal catalysis. **d** Conversion of PET and yield of BHET over $Li_{0.76}CoO_2$ for ten cycles. All error bars represent the standard deviations of three independent measurements and the bars indicate mean values.

system. The PET conversion and BHET yield remained analogous to those of thermal catalysis, thus ruling out the influence of the UV-activation process[35]. To further elucidate the enhanced photo-thermal catalytic performance, a kinetic analysis of PET glycolysis was undertaken in both photothermal and thermal catalytic settings (Supplementary Fig. 12). Generally, PET glycolysis conforms to a first-order reaction kinetics model, enabling the calculation of activation energy ($E_a$) through the Arrhenius formula (1).

$$\ln k = \ln A - \frac{E_a}{RT} \qquad (1)$$

where A, R and T refer to the pre-exponential factor, gas constant (8.314 J k⁻¹ mol⁻¹) and reaction temperature in Kelvin, respectively. The $E_a$ was determined to be 78.7 and 79.4 kJ mol⁻¹ for photothermal and thermal catalytic glycolysis, respectively (Fig. 3c). These values notably fall below the typical reported range (> 120 kJ mol⁻¹)[35], indicating a high catalytic ability of $Li_{0.76}CoO_2$. Notably, photothermal catalysis and thermal catalysis exhibited congruent activation energies, providing further affirmation that photothermal catalysis follows the same reaction pathway as thermal catalysis[36]. These observations reinforce the notion that the primary distinction between photothermal and thermal catalysis may be attributable to the localized heating effects[37].

We recognize the importance of evaluating the long-term stability of the catalyst comprehensively. The $Li_{0.76}CoO_2$ exhibited remarkable stability in photothermal catalysis, maintaining PET conversion rates and BHET yields at 98% and 96% of their initial performances even after ten catalytic cycles (Fig. 3d). We also carried out five long-term parallel experiments, aiming to simulate extended reaction periods. As the reaction time increases, the conversion rate of PET gradually rises. Given that the PET glycolysis reaction follows first-order kinetics, we conduct a linear fit of the data in Supplementary Fig. 13a, resulting in a

perfect linear relationship, as depicted in Supplementary Fig. 13b. This outcome indicates that the catalyst showcases excellent stability over prolonged reaction times.

Based on the aforementioned observations, we propose a pho-tothermal catalytic mechanism for the glycolysis of PET (Supplementary Fig. 14). Initially, $Li_{0.76}CoO_2$ converts sunlight to heat, facilitating the dissolution of PET into the EG solution. Subsequently, PET polymer chains migrate to the surface of the catalyst, where they are activated by $Co^{\delta+}$ species present on $Li_{0.76}CoO_2$. The activation process entails the coordination of $Co^{\delta+}$ with carbonyl oxygen atoms within PET chains, leading to charge delocalization towards oxygen and the generation of $C^{\gamma+}$ centers. Concurrently, the electron-rich oxygen atom within EG engages in a nucleophilic substitution reaction with the $C^{\gamma+}$ centers, resulting in the cleavage of the PET chain. It is plausible to infer that, after n-1 efficient cleavage events, a PET molecular chain with a polymerization degree of n can be depolymerized into n BHET monomers.

### Demonstration of photothermal glycolysis of PET

An outdoor experimental investigation was carried out to assess the practical viability of $Li_{0.76}CoO_2$ in the context of photothermal cata-lytic PET glycolysis under sunlight irradiation (Fig. 4a). Utilizing a poly(methyl methacrylate) Fresnel lens with a diameter of 110 cm as the concentrating lens, the solar irradiance at the focal point achieved an approximate value of 6.0 W cm⁻². As a result, the temperature of the reaction solution escalated to the boiling point of EG (197 °C) within a mere 50 s of irradiation. Despite minor fluctuations in sunlight inten-sity recorded during the experimental trial, the solution temperature exhibited remarkable stability, consistently maintaining a value of 197 °C (Fig. 4b). After 20 min of reaction time, PET flake conversion reached a complete 100%, ultimately yielding 42 g of BHET following the filtration and recrystallization steps (Fig. 4b).

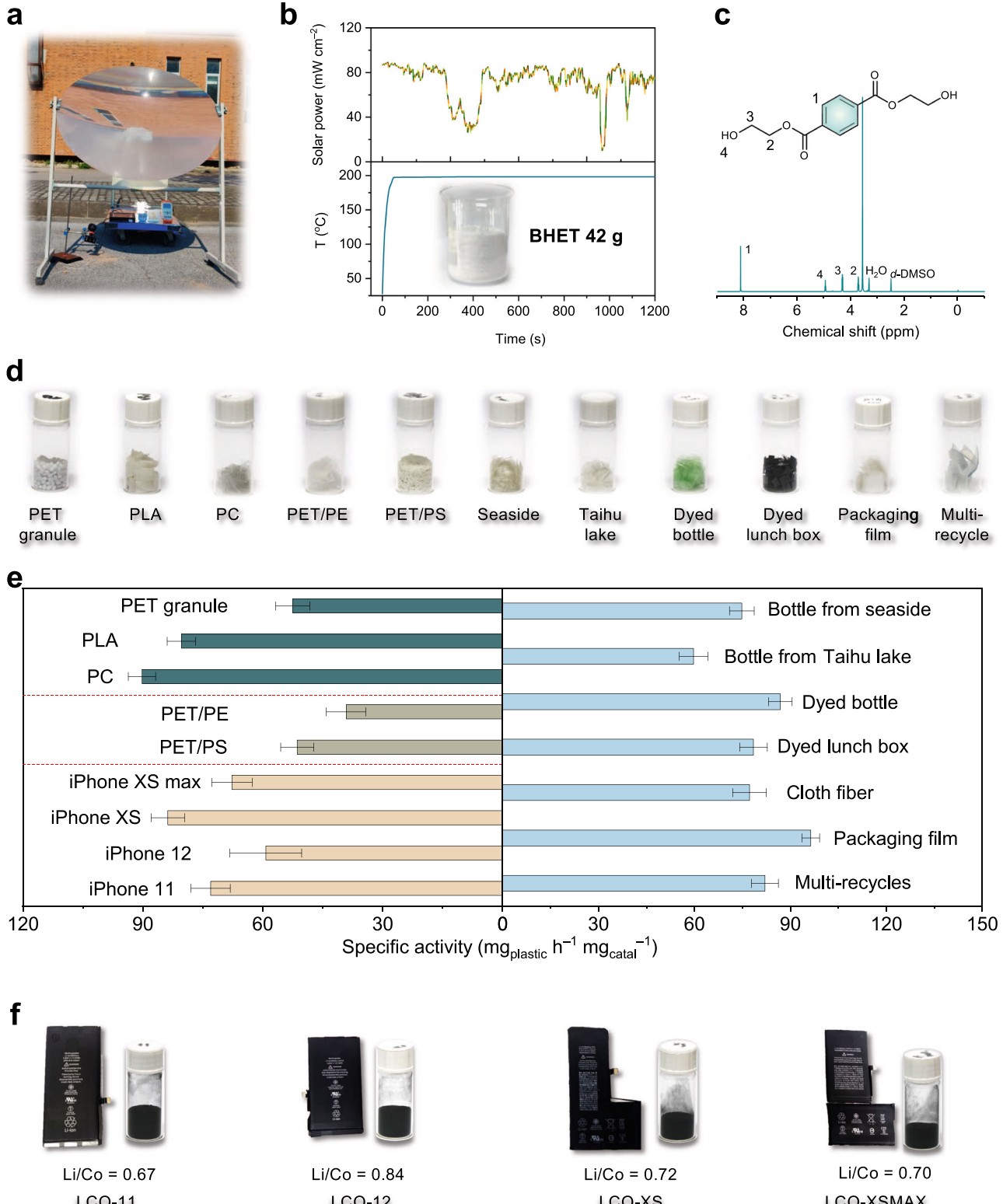

**Fig. 4 | Demonstrations of photothermal catalysis. a** Portable outdoor test system. **b** Real-time monitoring the outdoor experiment. Upper row: solar power; Bottom row: system temperature. **c** $^1$H NMR spectra of produced BHET with internal standard of dioxane. Inset is the digital image of 42 g of BHET. **d** Digital images of various real-world polyesters. **e** The specific activities of glycolysis of various polyesters. Green column: pure polyesters; Gray column: mixed plastics; Yellow column: glycolysis of PET over different real spent LCO powders; Blue column: real-world PET wastes. **f** Digital images of spent LCO batteries and powders, and corresponding Li-Co ratio. All error bars represent the standard deviations of three independent measurements and the bars indicate mean values.

Regarding to the challenges posed by natural fluctuations in sunlight intensity during outdoor testing, we designed and implemented a solar photothermal catalysis system with solar intensity tracking functionality and adjustable light intensity (Supplementary Fig. 15). The system comprises essential components, including a light tracking sensing and processing system, a light-focusing system, and a servo motor rotation system. By adjusting the azimuth angle of the focusing mirror, the system effectively tracks sunlight, stabilizing the incident light intensity during outdoor experiments. While our current setup is still in its early stages and the equipment is bulky, it is crucial to highlight that the temperature control error of the system is currently within ~7 °C. This demonstration underscores the substantial potential of $Li_{0.76}CoO_2$ nanoparticles in harnessing natural sunlight for the sustainable recycling of waste polyester.

The assessment of BHET purity was conducted through nuclear magnetic resonance (NMR) analysis, with the inclusion of 1,4-dioxane as an internal standard. As depicted in Fig. 4c, the singlet peak at 8.1 ppm corresponds to hydrogen atoms located on the benzene ring. The triplet peaks at 4.9 ppm, 4.3 ppm, and 3.7 ppm represent the hydrogen atoms of the terminal hydroxyl group and $CH_2$ groups, respectively. Additionally, the peak positions discerned in the $^{13}C$ NMR spectrum (Supplementary Fig. 16) further corroborate the exceptional purity of the BHET product. The purity of BHET was calculated as 99.5% according to the peak area ratio with internal standard (Supplementary Fig. 17), which meets the requirements of re-polymerization processes to close the recycling loop.

### Recycling of real-world waste polyesters

Efforts to recycle real-world plastic waste are of paramount importance in addressing the pervasive issue of plastic pollution, conserving valuable resources, and facilitating the shift toward a more sustainable and environmentally friendly approach to plastic utilization and disposal. In our research, we harnessed the catalytic capabilities of $Li_{0.76}CoO_2$ to facilitate the recycling of PET materials sourced from diverse origins (Fig. 4d). These sources included colored bottles, lunch boxes, fibers, mixed plastics, and multi-recycled plastics, among others. As depicted in Fig. 4e, even in the presence of impurities and additives commonly found in post-consumer PET products, the catalytic performance of $Li_{0.76}CoO_2$ remained robust and comparable to that of particles devoid of additives. Furthermore, plastic waste streams frequently comprise mixed polymers, posing an additional technical challenge. Nevertheless, $Li_{0.76}CoO_2$ consistently demonstrated remarkable efficacy when utilized in physical mixtures of PET with various plastics, including polystyrene (PS) and polyethylene (PE). Our research also showcased $Li_{0.76}CoO_2$ catalytic prowess across three distinct polyesters, achieving specific activities of 80.36 $mg_{PLA}$ $mg_{catal}^{-1}$ $h^{-1}$ and 90.23 $mg_{PC}$ $mg_{catal}^{-1}$ $h^{-1}$ for polylactic acid (PLA) and polycarbonate (PC), respectively. These findings underscore the suitability of $Li_{0.76}CoO_2$ for effectively managing real-world polyester wastes. The synthesized $Li_{0.76}CoO_2$ possesses readily accessible active sites, enabling the depolymerization of a wide range of substrates.

To evaluate the catalytic efficacy of spent LCO in photothermal catalytic polyester glycolysis, we initially procured four used iPhone batteries and subsequently acquired LCO powders (named LCO-11/12/XS/XSMAX, Fig. 4f) following the procedures outlined in Fig. 1c. The Li-Co ratio of these LCO powders fell within the range of 0.65–0.85 (Fig. 4f), determined through ICP-OES analysis (Supplementary Table 4). The XRD patterns obtained for these powders unequivocally confirmed their identity as LCO, with no discernible damage to the crystal structure or the presence of extraneous impurities (Supplementary Fig. 18). The specific activities of various spent LCO are summarized in Fig. 4e and Supplementary Fig. 19. While the catalytic properties of the four catalysts exhibit variations, they consistently demonstrate elevated catalytic activity and monomer yield in comparison to these of pristine LCO. These findings underscore the potential of spent LCO as a viable photothermal catalyst for polyester glycolysis.

### Techno-economic assessment

The systematic examination of a designated waste plastics recycling technology through the techno-economic assessment (TEA) stands as an imperative endeavor. In pursuit of this goal, we harnessed the capabilities of Aspen Plus V12 software to construct a comprehensive process model for the glycolysis of PET[38–40]. Supplementary Fig. 20 and Supplementary Table 5 provide process flow diagram and crucial stream parameters. Furthermore, we used Aspen Process Economic Analyzer V12 to meticulously evaluate both the capital outlay and operational costs associated with the simulated PET glycolysis plant (Supplementary Table 6). Employing discounted cash flow analysis, we computed the Minimum Selling Price (MSP) required to render PET glycolysis a viable process for the production of BHET[41]. To enhance the precision of our evaluation, we conducted sensitivity analysis on pivotal process parameters. This exercise effectively pinpointed the critical factors that exert a pronounced influence on the economic feasibility of the PET glycolysis process.

In our baseline scenario, we established several key assumptions to assess the practical production performance of the process. Firstly, we sourced the PET material for depolymerization from clean, colored sheets, setting the price at 0.66 \$ $kg^{-1}$, a value corroborated by existing literature references[42]. To gauge the process's real-world production capability, we configured it to depolymerize 11,905 kg of PET per hour, translating to an annual throughput of 100,000 tons. Subsequently, we implemented hot filtration technology to recuperate the catalyst for subsequent use, assuming in this scenario that the catalyst is replaced six times per year. The final stage involved the acquisition of BHET crystals through a cold filtration process, completing the recycling loop.

With the annual BHET production volume as a basis, we calculated the MSP, which stood at 1.135 \$ $kg^{-1}$. Notably, the primary cost contributor within the overall expenditure framework was the raw material expense for post-consumer PET, as elucidated in Fig. 5a and Supplementary Table 7. In our pursuit of a more nuanced understanding, we conducted a sensitivity analysis to assess the impact of fluctuations in the cost of post-consumer polyester raw materials on the MSP, as vividly depicted in Fig. 5b and Supplementary Table 8. Remarkably, as the price of PET raw materials exhibited a range of variation, from 0.22 \$ $kg^{-1}$ to 1.10 \$ $kg^{-1}$, the MSP for BHET demonstrated a corresponding increase, ascending from 0.901 \$ $kg^{-1}$ to 1.342 \$ $kg^{-1}$. Furthermore, we meticulously scrutinized uncertainties associated with factors including EG cost, catalyst raw material pricing, catalyst replacement frequency, and factory scale, as encapsulated in Supplementary Figs. 21–24 and Tables 9–11. Within this context, it is noteworthy that, owing to the low content and high recyclability of the catalyst, fluctuations in catalyst cost and replacement frequency exerted a negligible influence on the overall MSP for recycled BHET.

Through the systematic simulation of various scenarios, we have successfully aggregated uncertainties to discern the principal drivers exerting influence over the MSP of recycled BHET, as artfully illustrated in Fig. 5c and Supplementary Table 12. Amid the array of factors susceptible to uncertainties, it is abundantly clear that the cost of raw materials in the form of recovered post-consumer PET stands as the most formidable determinant of the MSP for recycled BHET. Moreover, our investigative efforts encompassed the examination of different factory scales and their impact on the MSP. Recognizing the market-driven nature of post-consumer polyester material costs, we strategically aimed to enhance profitability through factory scaling, as succinctly depicted in Fig. 5d and Supplementary Table 13. It is noteworthy that as the factory's annual throughput exceeds 10,000 tons of post-consumer polyester, the MSP for recycled BHET descends below

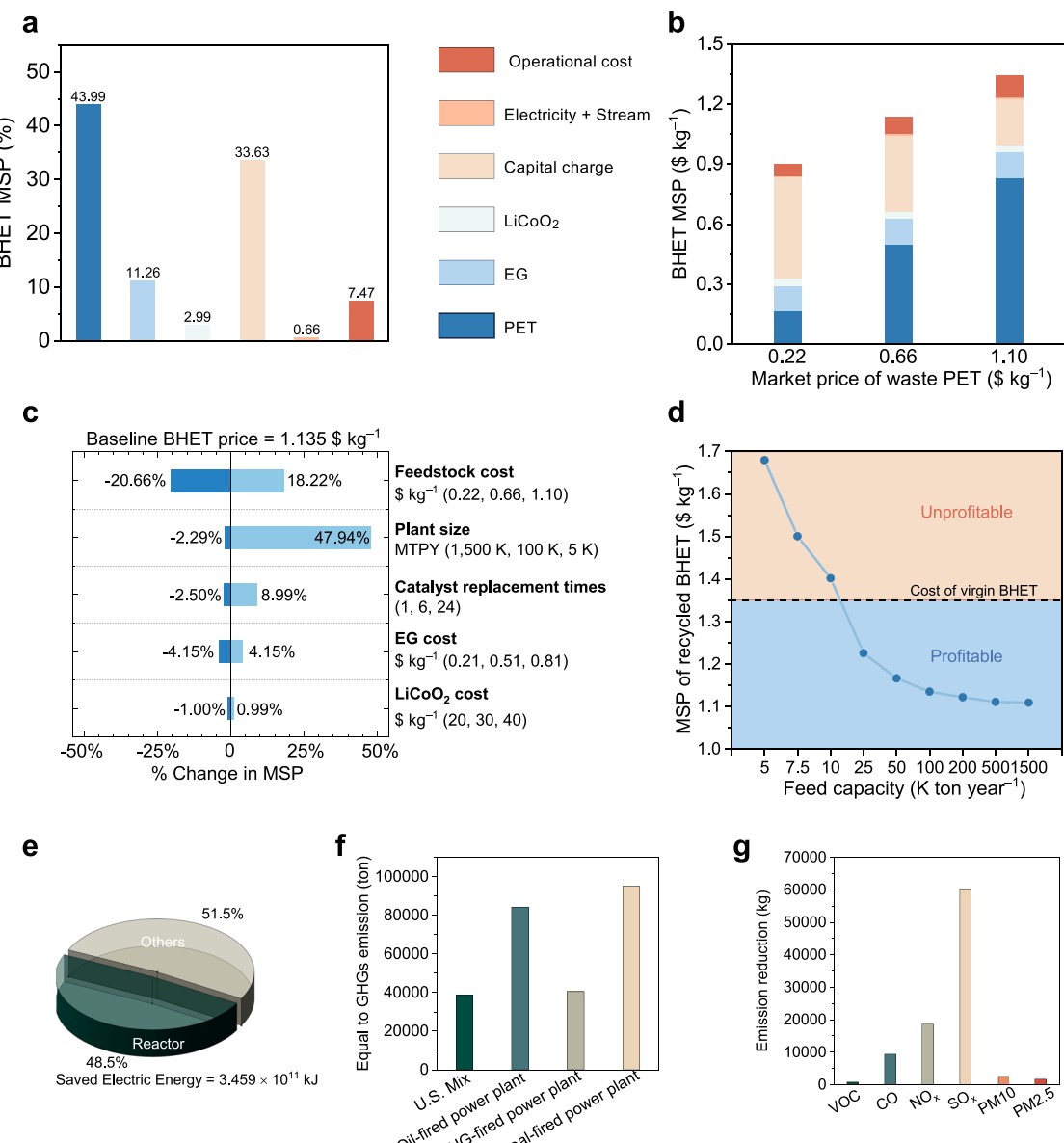

**Fig. 5 | TEA, energy and environment impact of photothermal catalysis. a** Cost breakdown of the BHET MSP in the base case process and MSP with percentage contribution of each factor. **b** Cost breakdown of the BHET MSP in the base case process design and as a function of PET feedstock price. **c** Tornado plot summarizing the effect of different process variables investigated on the BHET MSP. **d** BHET MSP as a function of process variables of MTPY. **e** Simplified estimation of energy consumption in different equipment for recycling 100,000 tons of PET. **f**–**g** The impact of photothermal catalysis on reducing GHGs emission (**f**) and gaseous pollutants emission (**g**) compared to thermal catalysis.

the average market price of virgin BHET (1.35 \$ kg$^{-1}$), signifying a financially favorable position. Conversely, in the scenario with an annual processing capacity of 5000 tons of post-consumer polyester, the MSP for recycled BHET reaches 1.679 \$ kg$^{-1}$, marking a notable 47.93% increase compared to the base case scenario. Additionally, in conjunction with the two primary influencing factors delineated above, it is imperative to acknowledge the impact of the remaining factors, which, in decreasing order of significance, encompass the cost of EG, catalyst replacement frequency, and catalyst cost.

We calculated the energy consumption based on recycling 100,000 tons of PET waste plastic, revealing that the reactor's energy consumption constitutes 48.5% of the total process (Fig. 5e). This implies that photothermal catalysis can save this proportion of energy. To provide more detailed insights, photothermal catalysis can conserve 3.459 × 10$^{11}$ kJ of electrical energy for every 100,000 tons of PET

recycled. The resultant reduction in greenhouse gas (GHG) emissions is equivalent to offsetting 38,716 tons of emissions from U.S. blended combustion power generation (Fig. 5f). In addition, recycling 100,000 tons of PET through photothermal catalysis can decrease acidic gas emissions by 93.60 tons (Fig. 5g). In summary, photothermal catalysis for plastic recycling is advantageous compared to traditional thermal catalysis systems in terms of efficiency, energy consumption, greenhouse gas emissions, and recycling costs.

## Discussion

In summary, we have proposed a novel upgrading route to discarded LIBs, wherein the extraction of the battery's cathode material serves as a photothermal catalyst for the recycling of various waste polyesters. Under simulated sunlight irradiation at 0.82 W cm$^{-2}$ for 30 min, we achieved a PET conversion rate of 96.34%, boasting a purity exceeding 99.5%. Moreover, the catalyst demonstrated strong stability

throughout ten recycling trials. Further outdoor experiments, assessments of energy consumption, and analyses of environmental impact all underscore the efficacy and eco-friendliness of PET recycling with retired batteries. An economic analysis indicates that the MSP for BHET produced via PET upcycling is 1.135 \$ $kg^{-1}$, which is significantly lower than the market price of BHET (1.60–1.98 \$ $kg^{-1}$).

The core inspiration behind the work presented here draws from the foresight of Buckminster Fuller's statements: "There is no energy crisis, just a crisis of ignorance." This visionary concept serves as the foundation for our meticulously crafted photothermal catalysts derived from spent LIBs. Our future research endeavors will be centered on an exhaustive exploration of the impact of light on catalytic reactions. We aim to introduce additional photochemical activation to further enhance the catalytic efficiency and product selectivity. Our study serves as a testament to the immense potential inherent in efficient spent LCO catalysts, exclusively driven by the focused power of natural sunlight. Catalysts exhibiting heightened reactivity, solely fueled by sunlight, hold substantial promise as a technology capable of leaving a negative carbon footprint. Furthermore, it is essential to acknowledge the diversity of used batteries. Furthermore, exploration of high-value utilization for lithium iron phosphate (LiFePO$_4$) and nickel manganese cobalt oxides (NMC) cathodes is necessary and important as well.

## Methods

### Materials
PET film (thickness = 25 μm), PET granule, PC, PET/PE, PS and multi-recycled PET were purchased from Alibaba Group. LiCoO$_2$ was purchased from Canrd. EG, K$_2$S$_2$O$_8$, N-Methylpyrrolidone (NMP), sodium chloride (NaCl), dimethyl carbonate (DMC) and dioxane were purchased from Sigma-Aldrich. Dimethyl sulfoxide (d-DMSO) was purchased from Energy Chemical. The spent LIBs (iPhone 11, iPhone 12, iPhone XS and iPhone XS Max) were collected from the electronic market.

### Cathode material harvesting
We collected discarded LIBs from the electronic marketplace and conducted initial processing before formal experimentation. To minimize the risk of combustion or explosion during the dismantling process, we immersed the used batteries in a saturated NaCl solution for 24 h to ensure complete discharge of any remaining charge. Afterward, we dried the fully discharged batteries in a vacuum oven at 60 °C for 8 h before manually disassembling their components within a glovebox. The cathode strips underwent a thorough rinsing procedure using DMC to remove residual electrolytes. Following thorough drying, the cathode strips were immersed in NMP for 30 min and then sonicated for 20 min, effectively removing LCO powders, binder material, and carbon black from the aluminum substrates. Subsequently, we subjected the resulting suspension to centrifugation separation (1360 g × 5 min).

The pristine LCO powder (5.0 g) underwent a ball milling process using a Fritsch planetary ball milling machine (Model P-6), operated continuously for 5 h at a rotational speed of 600 rpm. The milling process took place in a zirconia pot with an inner volume of 50 ml, utilizing zirconia balls with a diameter of 2 mm. To maintain optimal milling conditions, we maintained a weight ratio of 1:20 between the sample and the zirconia balls throughout the milling procedure. Following milling, the resulting product was washed with H$_2$O three times. The final product obtained from this process amounted to 4.8 g of Li$_{0.92}$CoO$_2$.

### Chemical delithiation of Li$_{0.92}$CoO$_2$
The 40 g of K$_2$S$_2$O$_8$ was dissolved in 1.0 L of distilled water at 50 °C to prepare a 0.15 M potassium persulfate solution. Next, 0.4893 g of Li$_{0.92}$CoO$_2$ powder was added into 25 mL of the persulfate

solution. The suspension was continuously stirred at 50 °C for various durations. Upon completion, the products were centrifuged out at 7104 g for 3 min, followed by a triple washing with distilled water and drying.

### Electrochemical delithiation of Li$_{0.92}$CoO$_2$
Initially, the assembled LCO coin cells were charged at a rate of 0.1 C with a cutoff voltage of 4.3 V. The lithiation degree of the charged LCO cathode was determined to be 0.56 by ICP measurement. To achieve specific lithiation degrees of 0.92, 0.76, and 0.60 for LCO, cutoff voltages of 3.93, 3.97, and 4.08 V were set, respectively. The lithium contents measured in the charged LCO cathodes closely align with the predefined values.

### Photothermal conversion ability
We introduced 2.5 g of EG and 10 mg of the catalyst into a custom-made reactor equipped with an inserted thermocouple for temperature monitoring. Subsequently, the reactor was positioned under simulated sunlight (CEL-HXF300-T3, China Education Au-light company) with a light intensity of 0.82 W $cm^{-2}$ for a duration of 30 min. Throughout this exposure, the solution temperature was continuously monitored and recorded by the thermocouple. For the photothermal stability test, the experimental conditions mirrored those described above. However, in this case, when the solution temperature reached 190 °C, the light source was promptly turned off, and the system was allowed to naturally cool down to 50 °C. This operation was repeated four additional times, ensuring rigorous testing of the photothermal stability under the specified conditions.

### Photothermal catalysis
In a homemade reactor equipped with an inserted thermocouple, we combined 0.5 g of PET film, 2.5 g of EG, and 10 mg of the catalyst. Subsequently, the reactor was positioned under simulated sunlight while maintaining a constant light intensity of 0.82 W $cm^{-2}$ and a fixed reaction duration, with continuous stirring at 200 rpm. After the depolymerization process, PET and its oligomers, along with the catalyst, were separated via filtration. The resulting mixture was carefully weighed, and the mass of the catalyst was subtracted to determine the remaining PET mass. The resulting filtrate consisted of a mixture comprising residual EG, monomers, and water. BHET was selectively isolated through thermal filtration. To ensure complete crystallization of BHET, the filtrate was placed in a refrigerator at 4 °C for a period of 10 h. Subsequently, a second round of filtration was performed to separate the BHET crystals. The collected BHET cake was then dried in a 70 °C oven for 5 h and weighed.

Please refer to the Supplementary Note One for detailed information of characterization, control experiments (Supplementary Tables 14, 15), stability tests, and economic and environmental analysis.

## Data availability
The source data generated in this study are provided in the Source Data file. Source data are provided with this paper.

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

## Acknowledgements

This work was supported by the National Natural Science Foundation of China (22376152 to J.C., 22208365 to P.X.), National Key Research and Development Program of China (2023YFC3903200 to J.C.), Natural Science Foundation of Jiangsu Province (BK20220298 to P.X.), Gusu Innovation and Entrepreneurship Leading Talent Program (ZXL2022492 to J.C., ZXL2022463 to P.X.), the Suzhou Frontier Technology Research Advanced Materials Project (SYG202305 to J.C.), and the Youth Promotion Association of Chinese Academy of Sciences (2023000079 to P.X.). J.C. thanks the support from the Suzhou Key Laboratory of Advanced Photonic Materials and Suzhou Key Laboratory of Functional Nano & Soft Materials, Collaborative Innovation Center of Suzhou Nano Science & Technology, the 111 Project. The authors thank SSRF (beamlines 11B) for the allocation of synchrotron beamtime.

## Author contributions

Conceptualization: J.C.; Methodology: X.L., P.Y., B. J. and L.W.; Investigation: X.L., Q.L., L. Z., M.C. and P.X.; Writing – original draft: J.C. and X.L.; Writing – review & editing: J.C., M.C., G.W., Z.C. and Q.Z.; Funding acquisition: P.X. and J.C.; Supervision: J.C.

## Competing interests
The authors declare no competing interests.
