## [Peer Review File · Nature Communications]

Grave-to-Cradle Photothermal Upcycling of Waste Polyesters over Spent LiCoO₂REVIEWER COMMENTS

Reviewer #1 (Remarks to the Author):

The paper submitted by Jinxing Chen et al. focuses on the upcycling of LCO materials from end-of-life LIBs. Personally, I believe that critical raw materials contained in LIBs should be recovered for the production of new batteries, given their urgent demand. In contrast, the authors propose the application of LCO in a new context, testing it as a photocatalyst in polyester degradation. Setting aside this personal general comment, I believe the following issues need to be addressed before the manuscript can be considered for publication:

The cathode materials of end-of-life lithium-ion batteries (LIBs) may exhibit varying degrees of lithiation, depending on the state of charge at the time of disposal. This variability in lithiation levels implies that the suggested chemical delithiation method may not be universally applicable, given the diverse and heterogeneous lithiation degrees of that the LCO materials collected from end-of-life batteries. While this concern could be of secondary importance if the primary objective of the study was solely to demonstrate the catalytic properties of materials, the authors also propose a Techno-economic assessment and assign an economic value to the recovered catalyst. Consequently, the delithiation method should, even in principle, be scalable to be considered in a techno-economic assessment and this seems to not be the case. In addition, the authors should provide clarification on the process for achieving a lithiation degree higher than what is eventually found in waste, especially when it is lower than the optimal range for catalytic performance. A lithiation step could be needed.

Moreover, concerning the selection of the lithiation degree for the evaluation of catalyst properties, it appears that a systematic and rational approach was not employed. Instead, it seems that the authors tested the LCO cathode resulting from various durations of chemical delithiation without a predefined target for the lithiation degree. In this view, since the reported crucial role of the lithiation degree a more controllable method should be considered such as, for example, electrochemical delithiation (<https://doi.org/10.1016/j.jechem.2023.09.040>).

The method by which the authors separate the conductive carbon from the LCO cathode, obtaining an almost pure LCO material, is not clearly explained. Furthermore, it raises the question of whether all necessary separations required to recover pure LCO from the battery waste have been accounted for in the techno-economic assessment.

Reviewer #2 (Remarks to the Author):

I appreciate the author's efforts toward the sustainable utilization of Co-based batteries as catalysts to break the polymers into monomers. How impactful is using cobalt as a catalyst? It was recognized as a conflict metal and involved unethical mining activities.

1. Authors need a thorough understanding of current trends in the battery market and usage of Co-based batteries. They mentioned LCO battery usage of more than 70%. However, Cobalt is considered a conflict metal and it's almost a half-decade now. It's majorly focused on LiFePO₄, NMC, and high nickel-based cathodes (<https://iea.blob.core.windows.net/assets/dacf14d2-eabc-498a-8263-9f97fd5dc327/GEVO2023.pdf>).
2. Authors didn't study why the Li_{0.76}CoO₂ showed factor of 10.24 higher than LiCoO₂? It needs a thorough investigation of their explanation of catalytic activity.
3. The authors mentioned the effect of temperature on the catalytic activity. We know the Sun never shines all the time and how to control the temperature. Does this affect the cost of recycling or have authors considered it?
4. Can Authors compare this photocatalysis with other sustainable processes of recycling polymers?

The mentioned cost analysis considered battery recycling but not the other polymer recycling process.

5. These failed batteries won't contain single-phase LiCoO_2 , and the authors also mentioned it. Then it may need a thorough analysis of which specific composition works better and why. It may have a huge effect the techno-economics if used catalyst activity is poor.
6. Catalyst poisoning is a big issue in metal-based catalysis. Did the authors perform any long-term tests on this catalyst? How often does it need to change or how quickly reduce its efficacy?
7. Table S4 needs a fix.

Reviewer #1:

The paper submitted by Jinxing Chen et al. focuses on the upcycling of LCO materials from end-of-life LIBs. Personally, I believe that critical raw materials contained in LIBs should be recovered for the production of new batteries, given their urgent demand. In contrast, the authors propose the application of LCO in a new context, testing it as a photocatalyst in polyester degradation. Setting aside this personal general comment, I believe the following issues need to be addressed before the manuscript can be considered for publication.

Response: We sincerely appreciate your recognition of the novelty of our work. We agree with your viewpoint regarding the potential recovery of cobalt from spent LCO batteries for the production of new batteries. Additionally, cobalt exhibits significant role in various applications, as outlined in Fig. R1, such as in the field of catalysis. Recently, the escalating concern over waste plastic pollution, with a global accumulation exceeding 10.5 billion tons by 2020, has evolved into a pressing environmental crisis. In this work, we demonstrated that spent LCO can be repurposed as an effective catalyst for plastic recycling, yielding valuable products. This not only diminishes the carbon footprint but also contributes substantially to resource sustainability in both battery and plastic industries. Of particular note is the minimal quantity of cobalt catalyst required for plastic recycling, which has a negligible impact on the battery recycling and manufacturing market. Our dedicated efforts here are centered on diversifying possible applications for cathode materials derived from spent LIBs.

Fig. R1 Distribution of cobalt demand worldwide in 2022 by applications.

(<https://www.statista.com/statistics/1143399/global-cobalt-consumption-distribution-by-application/>)

1. The cathode materials of end-of-life lithium-ion batteries (LIBs) may exhibit varying degrees of lithiation, depending on the state of charge at the time of disposal. This variability in lithiation levels implies that the suggested chemical delithiation method may not be universally applicable, given the diverse and heterogeneous lithiation degrees of that the LCO materials collected from end-of-life batteries. While this concern could be of secondary importance if the primary objective of the study was solely to demonstrate the catalytic properties of materials, the authors also propose a Techno-economic assessment and assign an economic value to the recovered catalyst. Consequently, the delithiation method should, even in principle, be scalable to be considered in a techno-economic assessment and this seems to not be the case. In addition, the authors should provide clarification on the process for achieving a lithiation degree higher than what is eventually found in waste, especially when it is lower than the optimal range for catalytic performance. A lithiation step could be needed.

Response: Thank you for your valuable suggestions. Fundamentally, chemical delithiation can serve as a valuable method for a thorough exploration of the relationship between the Li content in LCO and its catalytic performance. Through precisely controlling the lithiation degree in LCO, our study reveals that the valence state and electronic structure of cobalt significantly impact catalytic efficiency and monomer yield. The optimal Li/Co ratio in LCO for PET glycolysis was determined to be approximately 0.76 (Fig. R2), which can provide clear guidance for practical screening of spent LIBs.

In practical applications, we intend to select proper spent batteries through the assessment of open circuit voltage (OCV), a parameter directly indicative of the Li content in the cathode (*Nat. Commun.* **2014**, *5*, 5381; *Nat. Energy* **2021**, *6*, 781-789; *Nat. Energy* **2018**, *3*, 373-386). Given the limited demand for spent LCO batteries in catalyzing plastic recycling, the screening process will not be overly extensive, **rendering it both feasible and scalable in practical applications.**

Fig. R2 Conversion of PET and yield of BHET over Li_{1-x}CoO₂

When the lithiation degree in spent LIBs falls below the optimal value, a relithiation step might be necessary, as suggested by the reviewer. In the field of battery recycling, relithiation technologies

have seen significant progress, with various methods developed, including solid-state relithiation, chemical relithiation, ionothermal relithiation, hydrothermal relithiation, and molten salt relithiation. For obtaining LCO with a predefined Li composition, solid-state relithiation with precise control of Li salt addition would be preferable. This process is straightforward and holds great potential for practical applications. **We have incorporated your suggestions into our revised manuscript.**

2. Moreover, concerning the selection of the lithiation degree for the evaluation of catalyst properties, it appears that a systematic and rational approach was not employed. Instead, it seems that the authors tested the LCO cathode resulting from various durations of chemical delithiation without a predefined target for the lithiation degree. In this view, since the reported crucial role of the lithiation degree a more controllable method should be considered such as, for example, electrochemical delithiation (<https://doi.org/10.1016/j.jechem.2023.09.040>).

Response: Thank you for your constructive comments. Following the reviewer's suggestions, we implemented an electrochemical delithiation method to regulate the lithiation degree of LCO. Initially, the assembled LCO coin cells were charged at a rate of 0.1C with a cutoff voltage of 4.3 V (**Fig. R3a**). The lithiation degree of the charged LCO cathode was determined to be 0.56 by ICP measurement. The correlation between charging voltage and lithiation degree is illustrated on the upper x-axis in **Fig. R3a**. To achieve specific lithiation degrees of 0.92, 0.76, and 0.60 for LCO, we set cutoff voltages of 3.93, 3.97 and 4.08 V (**Figs R3b–R3d**). The lithium contents measured in the charged LCO cathodes closely align with the predefined values (**Table R1**).

Subsequently, we explored the electrochemically delithiated LCO samples for catalyzing PET glycolysis (**Fig. R4**). Our findings revealed that the catalytic activity was most pronounced for the LCO sample with a lithiation degree of 0.72, closing to the results observed in chemically delithiated samples, suggesting a substantial influence of the LCO composition on the catalyst's activity. This outcome underscores the effectiveness of the electrochemical delithiation method in regulating the lithiation degree. **These discussions have been incorporated into the revised manuscript.**

Fig. R3 The electrochemical delithiation process. Charge curve at 0.1C with the cutoff voltage of 4.30 V (a), 3.93 V (b), 3.97 V (c) and 4.08 V (d).

Fig. R4 Conversion of PET glycolysis and yield of BHET over LCO-X and pristine LCO catalysts. Commercial PET film was cut into small fragments (0.5 g, 0.5 × 0.5 cm) and immersed in a 2.5 g EG solution with 10 mg of the photothermal catalyst.

Table R5

analysis of C-LCO, LCO-1, LCO-2 and LCO-3.

Sample	Li (wt.%)	Co (wt.%)	Li/Co (mol/mol)
C-LCO	3.12	47.34	0.56
LCO-1	4.32	39.49	0.93
LCO-2	4.28	50.45	0.72
LCO-3	3.85	49.99	0.65

3. The method by which the authors separate the conductive carbon from the LCO cathode, obtaining an almost pure LCO material, is not clearly explained. Furthermore, it raises the question of whether all necessary separations required to recover pure LCO from the battery waste have been accounted for in the techno-economic assessment.

Response: Thank you for your suggestions. The removal of conductive carbon aligns with established methodologies outlined in previous reports (*Joule* **2020**, *4*, 2609–2626; *Nat. Sustain.* **2023**, *6*, 797–805; *J. Am. Chem. Soc.* **2022**, *144*, 20306–20314; *Adv. Mater.* **2023**, doi: 10.1002/adma.202311553). After safely disassembling spent LCO batteries, the cathode was immersed in NMP solvent. Through subsequent washing and centrifuging steps, the conductive carbon was successfully removed. In addition, our techno-economic analysis (TEA) has included necessary separation processes, referring to those provided in the EverBatt model (**Fig. R5**). The total cost for LCO separation is calculated to be \$1.05 per kilogram. These details have been incorporated into our revised manuscript.

Fig. R5 Process diagram of a generic recycling process.

Reviewer #2:

I appreciate the author's efforts toward the sustainable utilization of Co-based batteries as catalysts to break the polymers into monomers. How impactful is using cobalt as a catalyst? It was recognized as a conflict metal and involved unethical mining activities.

Response: Thank you for recognizing the novelty and importance of our work. Catalysts play a pivotal role in a myriad of applications, serving as essential agents that accelerate and facilitate chemical reactions. The escalating issue of waste plastic pollution, accumulating to over 10.5 billion tons globally by 2020, has become a critical environmental issue. Cobalt-based catalyst have been demonstrated to be effective in converting waste plastics to valued chemicals (*Adv. Funct. Mater.* **2023**, 33, 2210283; *JACS Au* **2022**, 2, 2259–2268), offering a promising solution to reduce the carbon footprint. **In this work, we innovatively repurposed spent LCO cathodes to catalyze plastic recycling, which reduces necessity for cobalt mining and refining activities, minimizing environmental damage and promoting resource sustainability. Our analysis shows that with the use of a small amount of Co-containing catalysts in photothermal catalysis, we can reduce the carbon emission of plastic recycling by 48.5%.**

1. Authors need a thorough understanding of current trends in the battery market and usage of Co-based batteries. They mentioned LCO battery usage of more than 70%. However, Cobalt is considered a conflict metal and it's almost a half-decade now. It's majorly focused on LiFePO₄, NMC, and high nickel-based cathodes (<https://iea.blob.core.windows.net/assets/dacf14d2-eabc-498a-8263-9f97fd5dc327/GEVO2023.pdf>).

Response: Thank you for your valuable suggestions. Due to the cobalt shortage, the LIB industry is actively working to minimize cobalt content. Cathode materials like LiFePO₄, NMC, and high nickel-based compositions with low or no cobalt have gained prominence. However, it is crucial to acknowledge that cobalt plays a vital role in various applications, particularly in catalysis (**Fig. R1**). **Recycling cobalt resources for use in other applications can help mitigate the need for additional cobalt mining and refining activities, significantly contributing to reducing the CO₂ footprint. In this context, we used discarded LCO as a catalyst for polyester upcycling, addressing environmental concerns while also enhancing the sustainability of critical materials.**

Furthermore, exploration of high-value utilization for LiFePO₄, NMC, and high nickel-based cathodes is necessary as well. We are planning to expand our current system to incorporate other spent cathodes in our initiatives focused on plastic recycling. This extension is aimed at broadening the scope of our research and further contributing to the development of sustainable solutions for plastic

recycling. We express our gratitude once again for the constructive suggestions.

Fig. R1 Distribution of cobalt demand worldwide in 2022 by applications.

(<https://www.statista.com/statistics/1143399/global-cobalt-consumption-distribution-by-application/>)

2. Authors didn't study why the $\text{Li}_{0.76}\text{CoO}_2$ showed factor of 10.24 higher than LiCoO_2 ? It needs a thorough investigation of their explanation of catalytic activity.

Response: Thank you for your valuable suggestions. According to our previous report, the catalytic activity for the glycolysis of PET is strongly related to the Co–O₆ unit in LCO (*Adv. Funct. Mater.* **2023**, *33*, 2210283). Throughout the catalytic process, $\text{Co}^{\delta+}$ activates the carbonyl oxygen atom of the PET main chain, inducing charge delocalization towards oxygen and generating C⁺ centers. Simultaneously, the oxygen atom in EG attacks the C⁺ center, initiating a nucleophilic substitution reaction. This sequence leads to the breakage and depolymerization of the PET chain (**Fig. R6**). The XANES spectra presented in Fig. R7 illustrate noticeable changes in electronic structure of cobalt as the degree of delithiation in LCO increases. Specifically, the delithiation of LCO results in an increase in the valence state of the Co element ($\text{Co}^{\delta+}$, $\delta > 3$), which induces the contraction of the Co–O₆ unit cell (**Fig. R8**, *Adv. Funct. Mater.* **2020**, *30*, 2002223; *ACS Energy Lett.* **2023**, *8*, 4806–4817). Consequently, the activation ability of $\text{Co}^{\delta+}$ toward carbonyl oxygen in PET is stronger than that of Co^{3+} in LiCoO_2 , resulting in an enhanced catalytic performance. These discussions have been incorporated into the revised manuscript, and we trust that these additions enhance the clarity and depth of our work.

Fig. R6 Integrated functionalities of $\text{Li}_{0.76}\text{CoO}_2$ and the photothermal catalytic mechanism of PET glycolysis over $\text{Li}_{0.76}\text{CoO}_2$.

Fig. R7 Co K-edge XANES spectra of LiCoO_2 , $\text{Li}_{0.92}\text{CoO}_2$, $\text{Li}_{0.76}\text{CoO}_2$ and Co foil.

Fig. R8 Schematic views of the atomic structures of LiCoO_2 and $\text{Li}_{1-x}\text{CoO}_2$. The green, red, and gray spheres represent Co, O and Li atoms, respectively.

3. The authors mentioned the effect of temperature on the catalytic activity. We know the Sun never shines all the time and how to control the temperature. Does this affect the cost of recycling or have authors considered it?

Response: Thank you for your valuable suggestions. In laboratory test conditions, we employed a xenon lamp to simulate sunlight, enabling precise control of the reaction temperature through lamp current adjustments. For outdoor experiments, we implemented a solar-thermal catalysis system with solar intensity tracking functionality, as illustrated in **Fig. R9**. This system includes essential components such as a light-tracking sensing and processing system, a light-focusing system, and a servo motor rotation system. Through the adjustment of the azimuth angle of the focusing mirror, the system effectively tracks sunlight, ensuring a stable incident light intensity during outdoor experiments. Currently, the temperature control error of the system is within approximately 7 °C, meeting the requirements for the outdoor test. The entire light-tracking system is composed of low-cost stuff, which has little effect on the overall operation cost. While large-scale operational data will provide a more reliable cost analysis, a simple but conservative analysis based on an operational scale of 1000 kg per year would produce a \$1400 value product per year. Assuming the equipment would operate for 8 years, the equipment cost is only \$60 per year, which accounts for a minimal ratio to overall revenue.

Fig. R9 The solar-thermal catalytic system with automatic sun-tracking and adjustable light intensity.

4. Can Authors compare this photocatalysis with other sustainable processes of recycling polymers? The mentioned cost analysis considered battery recycling but not the other polymer recycling process.

Response: Thank you for your constructive suggestions. We have made comprehensive comparisons between photothermal catalysis and conventional thermal catalysis.

In terms of catalytic performance, photothermal catalysis shows enhanced catalytic efficiency and monomer yield (Fig. R10). With the same apparent temperature, the PET conversion rate in photothermal catalysis is 3.1 times higher. Mechanism studies reveal that the prominent improvement in catalytic performance is primarily attributed to the localized photothermal effect.

Furthermore, we conducted a detailed comparison of energy consumption and environmental impact between photothermal catalysis and thermal catalysis. For every 100,000 tons of waste PET recycled, photothermal catalysis can save 3.459×10^{11} kJ of electricity compared to thermal catalysis. This reduction translates to a decrease in greenhouse gas emissions by 38,716 tons and acid gas emissions by 93.60 tons, respectively (**Fig. R11**).

In the context of costs, for every 100,000 tons of PET recycled, thermal catalysis requires 142.86 tons of LCO, whereas photothermal catalysis only requires 46.08 tons. This results in significant savings in catalyst costs, calculated as $(142.86 - 46.08) \times 1000 \times 30$, equaling US\$2,903,400. Additionally, the savings in electrical energy costs can be calculated as $(3.459 \times 10^{11}/3600) \times 0.0775$, totaling \$7,446,458. Consequently, when recycling 100,000 tons of PET, the overall cost savings with photothermal catalysis compared to thermal catalysis amount to US\$10,349,858.

Therefore, photothermal catalysis for plastic recycling is advantageous compared to traditional thermal catalysis systems in terms of efficiency, energy consumption, greenhouse gas emissions, and recycling costs. We have incorporated all these discussions into the revised manuscript.

Fig. R10 Conversion of PET and yield of BHET over $\text{Li}_{1-x}\text{CoO}_2$ in photothermal and thermal catalysis.

Fig. R11 The impact of photothermal catalysis on reducing GHGs emission (left) and gaseous pollutants emission (right) compared to thermal catalysis for recycling 100,000 tons of PET.

5. These failed batteries won't contain single-phase LiCoO₂, and the authors also mentioned it. Then it may need a thorough analysis of which specific composition works better and why. It may have a huge effect the techno-economics if used catalyst activity is poor.

Response: Thank you for your valuable suggestion. Our systematic study revealed that the primary factor influencing catalytic performance is the Li/Co ratio of the material, which affects the electronic structure of cobalt. The XANES spectra for the delithiated LCO illustrate that delithiation leads to an increase in the valence state of the cobalt (Co^{δ+}, δ > 3). This change induces the contraction of the Co–O₆ unit cell, consequently enhancing the efficiency of Co^{δ+} in activating carbonyl oxygen in PET. Furthermore, we selected four different real LCO batteries (iPhone-11, iPhone-12, iPhone-XS, and iPhone-XSMAX), which show lithiation degree of 0.67, 0.84, 0.72, and 0.70 respectively. We observed that the Li/Co ratio of LCO-XS is close to the optimized value (~0.72), which exhibited the best performance for PET glycolysis (Fig. R11). This finding further underscores the substantial impact of the Li/Co ratio on catalytic performance.

Fig. R11 Conversion of PET glycolysis and yield of BHET over spent LCO from iPhone and pristine LCO catalysts. The LCO-11, LCO-12, LCO-XS and LCO-XSMAX represents cathode materials obtained from iPhone-11, iPhone-12, iPhone-XS and iPhone-XSMAX, respectively.

6. Catalyst poisoning is a big issue in metal-based catalysis. Did the authors perform any long-term tests on this catalyst? How often does it need to change or how quickly reduce its efficacy?

Response: Thank you for your valuable suggestions. We have carried out long-term tests to evaluate the stability of catalysts. The relationship between the conversion rate and reaction time for long-term tests is shown in **Fig. R12a**. As the reaction time increases, the conversion rate of PET gradually rises. Given that the PET glycolysis reaction follows first-order kinetics, we conduct a linear fit of the data in **Fig. R12a**, resulting in a perfect linear relationship, as depicted in **Fig. R12b**. **This outcome indicates that the catalyst showcases excellent stability over prolonged reaction times.** We sincerely appreciate your guidance, and these additional experiments make a significant contribution to the comprehensive assessment of the long-term stability of the catalysts.

Fig. R12 The stability of $\text{Li}_{0.76}\text{CoO}_2$ with 70 hours of continuous catalytic testing. **a** The conversion rates of PET at different times. **b** The linear fitting of conversion rate versus reaction time.

7. Table S4 needs a fix.

Response: Thank you for your careful reviewing, and we have fixed this table in the revised Supplementary Information.

REVIEWERS' COMMENTS

Reviewer #1 (Remarks to the Author):

Comments provided in the first revision round were addressed, including additional experiments carried out to adjust the degree of lithiation by electrochemical methods. In my opinion, the paper can now be published

Reviewer #2 (Remarks to the Author):

There is currently an urgent necessity to refrain from utilizing cobalt as a catalyst or cathode materials, given its association with child labor in the extraction process. The author's perspective on utilizing cobalt from discarded batteries is intriguing and their responses to my queries were satisfactory.